# Angular Heterotopic Pregnancy: Successful Differential Diagnosis, Expectant Management and Postpartum Care

**DOI:** 10.3390/medicina57111207

**Published:** 2021-11-05

**Authors:** Gabija Didziokaite, Monika Vitaityte, Gerda Zykute, Virginija Paliulyte, Arturas Samuilis

**Affiliations:** 1Faculty of Medicine, Vilnius University, 03101 Vilnius, Lithuania; monika.vitaityte@mf.stud.vu.lt (M.V.); gerda.zykute@mf.stud.vu.lt (G.Z.); 2Center of Obstetrics and Gynecology, Faculty of Medicine, Institute of Clinical Medicine, Vilnius University, 08661 Vilnius, Lithuania; Virginija.Paliulyte@santa.lt; 3Center of Radiology and Nuclear Medicine, Faculty of Medicine, Institute of Biomedical Sciences, Vilnius University, 08661 Vilnius, Lithuania; Arturas.Samuilis@santa.lt

**Keywords:** heterotopic pregnancy, angular pregnancy, pregnancy after ovulation induction, expectant management, interstitial pregnancy, follow-up

## Abstract

Heterotopic pregnancy is a rare, difficult to diagnose and life-threatening pathology, which requires timely decisions made by an experienced multidisciplinary team. In this type of multiple pregnancy there are both intrauterine and ectopic pregnancies present. Its incidence increases in pregnancies conceived by assisted reproductive technology or in pregnancies with ovulation induction. This article presents an angular heterotopic pregnancy case in a 34-year-old multigravida. The patient was admitted on the 14th week of gestation due to abdominal pain on the left side with suspicion of heterotopic pregnancy. Transabdominal ultrasound and magnetic resonance imaging (MRI) were performed to confirm the diagnosis of heterotopic angular pregnancy in the left cornu of the uterus. Multidisciplinary team made a decision to keep monitoring the growth of both pregnancies by ultrasound while maternal vitals were stable. Due to intensifying abdominal pain, diagnostic laparoscopy was performed. No signs of uterine rupture were observed, and no additional surgical procedures were performed. Maternal status and ultrasonographic findings were closely monitored. The mass in the left cornu of the uterus did not change significantly and the fetal growth of the intrauterine pregnancy matched its gestational age throughout pregnancy. At the 41st week of gestation, a healthy female neonate was born via spontaneous vaginal delivery. The incidence rate of heterotopic pregnancy tends to grow due to an increased number of pregnancies after assisted reproductive technology and ovulation induction. It is important to always assess the risk factors. The main methods for diagnosing heterotopic pregnancies are ultrasonography and MRI. The main management tactics for heterotopic pregnancy include expectant management as well as surgical or medical termination of the ectopic pregnancy. Expectant management may be chosen as an option only in a limited number of cases, if the clinical situation meets the specific criteria. When applicable, expectant management may reduce the frequency of unnecessary interventions and help to prevent patients from its complications.

## 1. Introduction

Heterotopic pregnancy is a type of multiple pregnancy, in which both intrauterine and ectopic pregnancies are present. It is a rare, difficult-to-diagnose and life-threatening pathology [1]. Its estimated incidence is 1:10,000–50,000 in spontaneous pregnancies. It is also reported to be as high as 1 in 7000 when the pregnancy is a result of assisted reproductive technology and 1 in 900 in pregnancies with ovulation induction [2]. Angular heterotopic pregnancy is a rare type of heterotopic pregnancy, which was only recently defined as a separate type of heterotopic pregnancy [3]. Diagnostics and treatment of angular ectopic pregnancy are still challenging due to lack of clinical experience on this subject as well as a lack of published clinical case reports.

## 2. Case Report

A 34-year-old gravida 4, para 2 female presented to the Perinatology Centre for management of presumed heterotopic angular pregnancy located in the left cornu of the uterus. The patient had undergone ovarian stimulation. Her gestational age was 13w + 3d based on early ultrasound. A Non-Invasive Prenatal Test showed no pathology. The patient’s prior pregnancies included two full-term normal spontaneous vaginal deliveries and one ectopic pregnancy. The patient had a history of laparoscopic ovarian cystectomy, appendectomy, cholecystectomy, laparoscopic treatment of ovarian apoplexy and laparoscopic treatment of ectopic pregnancy in the left fallopian tube.

On admission, the patient’s vitals were stable. The patient reported episodic abdominal pain on the left side. Transabdominal ultrasound imaging revealed a hypoechogenic 3.05 × 3.08 cm size mass in the left cornu of the uterus, filled with fluid (without viable embryo), which, by evaluation of the blood flow, could have been related to the uterus. Figure 1.

An urgent Magnetic Resonance Imaging (MRI) was performed. A 36 × 20 × 36 mm size cystic mass with T2-hyperintense wall in the left cornual region was observed. An MRI scan also showed one more fetus inside the uterine cavity with placenta located on the left lateral wall. Figure 2 The diagnosis of a heterotopic angular pregnancy in the left cornu of the uterus was established. The patient was hospitalized for further observation.

During hospitalization, blood and urine tests’ results were within the normal range. A multidisciplinary team (MDT) decided to keep monitoring the growth of both pregnancies by ultrasound while maternal vitals were stable and noted that further management would be determined according to the clinical situation. On the third day of hospitalization the patient started to feel more severe pain in the hypogastric region, radiating to the back and the left groin. The MDT made a decision to perform a diagnostic laparoscopy. Laparoscopy confirmed heterotopic angular pregnancy in the left enlarged, swollen cornu of the uterus. The left ovary and fallopian tube were not damaged and there were no signs of uterine rupture. Figure 3 and Figure 4.

The postoperative period was uneventful. After the surgery the MDT decided not to perform any further surgical interventions and to keep monitoring maternal status as well as ultrasonographic findings. The patient‘s general condition improved; the pain subsided.

In 6 days, ultrasonography was repeatedly performed. The hypoechogenic mass on the left cornual region of uterus was observed; however, its size remained unchanged. Another fetal ultrasound was scheduled in a week and the patient was discharged for further outpatient care. The similar ultrasonographic view was observed during the following ultrasound scans at the 15th and 20th wk.

On the 22nd week of gestation, the woman was admitted to tertial level hospital complaining of pain in hypogastric and left iliac regions of the abdomen, provoked by physical exercise. Ultrasonography was performed, the remaining unchanged mass in the left cornu of the uterus was observed as well as an intraamniotic septum in the lower segment of the uterus, as shown in Figure 5 and Figure 6. The fetal growth of the intrauterine pregnancy was unaffected and matched its gestational age. Conservative treatment was chosen, the pain resolved and the patient was discharged in 2 days.

At the 27th, 30th, 35th and 37th weeks of gestation, the ultrasonography was performed to monitor any possible changes of the mass in the left uterine cornu—the mass was compressed and its size did not differ significantly.

At the 41st week of gestation, the patient was admitted to the Obstetrics department due to the spontaneous rupture of membranes—transparent amniotic fluid was observed. A healthy female neonate (weight 2850 g, height 50 cm) with Apgar scores of 9 at 1 min and 10 at 5 min was born via spontaneous vaginal delivery in occiput posterior position. No Oxytocin was used during the labor. On the first day after delivery, transabdominal ultrasound was performed. A compressed mass of 3.3 × 1.4 cm remained visible in the left cornu of the uterus. Figure 7 Moreover, an intensified blood flow was observed on the left corner of the uterus in comparison with the right uterine corner. The postpartum and postnatal periods were uneventful, and the patient was discharged together with her newborn 2 days later.

After the discharge, follow-up visits were arranged in an outpatient clinic. The further postpartum period was uneventful, and the patient did not have any specific complaints. The β human chorionic gonadotropin (β-hCG) blood tests were performed on the 1st, 7th, 14th and 30th days after delivery—in 30 days it decreased drastically from 3602 international units per liter (IU/L) to 1.78 IU/L accordingly. Table 1

The follow-up transvaginal ultrasonography was performed after one month. The remaining mass sized 1.67 × 0.56 cm with visually more intensive blood flow was observed. Figure 8.

## 3. Discussion

Heterotopic pregnancy is a rare, difficult to diagnose and life-threatening pathology and its incidence tends to grow due to an increased number of pregnancies after assisted reproductive technology and ovulation induction [4]. Other risk factors for heterotopic pregnancy include inflammatory bowel disease, use of intrauterine spirals, ovarian hyperstimulation syndrome and ectopic pregnancies in the past. [1].

The most common presenting signs of heterotopic pregnancy include abdominal pain, peritoneal irritation, enlargement of uterus (larger than expected for the intrauterine pregnancy), adnexal mass. According to some authors, vaginal bleeding is less common for heterotopic pregnancies in comparison with ectopic pregnancies [1].

Heterotopic pregnancy is categorized by localization of ectopic pregnancy. Most common localization of such pregnancy is the fallopian tube (isthmus, ampulla, fimbriae), other, less common types include: angular, intramural, interstitial, cervical ectopic pregnancies [3,5]. Angular heterotopic pregnancy is a very rare type of heterotopic pregnancy. Before 1981, the name of angular pregnancy had been used as a synonym for interstitial pregnancy [3,6]. Later it became necessary to separate these two pathologies and the criteria for angular pregnancy were defined. Currently, these criteria include an enlarged, asymmetrical uterus, lateral swelling of uterus with round ligament lateralization and placental detention in the corner of the uterus, all observed during surgery [3,6]. The main difference between the two pathologies is the localization of implantation. In angular pregnancy, implantation happens in the endometrium, while in case of interstitial pregnancy, the embryo is implanted in the intramural part of the fallopian tube. Due to this difference, it is necessary to diagnose whether the gestational sac has a relation to the endometrium [7]. If it is present, angular pregnancy is more likely, if it is not present the diagnosis of interstitial pregnancy is favored. Additionally, interstitial pregnancy can be suspected if a myometrial mantle measurement gives results of less than 5 mm, in addition to a gestational sac being separate from the endometrium [7,8].

It is important to always assess the risk factors, as 70% of women with diagnosed heterotopic pregnancy have at least one of them [1]. It is particularly important to thoroughly examine the patients who have been treated with assisted reproductive technology [9]. The main and the first modality of choice for diagnosing heterotopic pregnancies is ultrasonography. The diagnostic role of human chorionic gonadotropin concentration in heterotopic pregnancy is debatable. It is important to differentiate this pathology from corpus luteum cyst and hemorrhagic cyst [1]. Nevertheless, heterotopic pregnancies are not always successfully diagnosed by ultrasound due its limitations, such as operator dependence, as well as limited visibility of the area inspected due to bowel gas or patient’s body habitus. Therefore, ultrasound cannot totally exclude heterotopic pregnancy in some cases. In these situations, MRI may be helpful as it provides a higher soft tissue contrast, better anatomical localization and it is less dependent on skills of the performing specialist [6].

The choice of treatment for heterotopic pregnancy is individual in each case. Treatment strategy depends on the number of previous pregnancies, the condition of intrauterine pregnancy, the doctor’s experience and patient’s socioeconomic status [10]. After heterotopic pregnancy diagnosis has been established, expectant management can be chosen if the patient is in a stable condition and intrauterine pregnancy is developing normally. If the clinical situation changes and indications to terminate an ectopic pregnancy occur, one of several tactics can be chosen: surgical or medical [10,11]. Surgical treatment options include laparoscopy and laparotomy. The aim of medical treatment is to terminate the ongoing ectopic pregnancy. For this purpose, transvaginal ultrasound-guided potassium chloride, hyperosmotic glucose or methotrexate injections into the gestational sac can be performed [12]. In many publications, laparoscopy is the first-line surgical treatment of ectopic pregnancy. Laparoscopic surgery causes less damage to the abdominal wall, decreases blood loss, minimizes the risk of postoperative complications, reduces the time of hospitalization and allows more rapid healing. Laparoscopic surgery also leads to a decreased risk of preterm labor and miscarriage because it involves less manipulation of the uterus [4]. Laparotomy is a better option to control the major intra-abdominal hemorrhage [13]. According to the analysis of 64 cases of heterotopic pregnancy, published in 2016, the rate of intrauterine fetal deaths was higher in the cases where surgical management was chosen—in this group, spontaneous miscarriage occurred in 14.8% cases [14].

Viable intrauterine pregnancy is an absolute contraindication for systemic methotrexate therapy; therefore, this treatment option was not appropriate for our case. Another possible treatment is a transvaginal ultrasound-guided potassium chloride, hyperosmotic glucose or methotrexate injection into the gestational sac [15]. Direct injection of methotrexate is believed to destroy the embryo and cause necrosis of the surrounding trophoblastic tissue [9]. However, the probability of methotrexate treatment failure is approximately 35% [16].

Expectant management may be chosen as an option if the patient does not experience severe symptoms, the embryo of the ectopic pregnancy has a limited craniocaudal length, no cardiac activity is registered and the level of β-hCG is decreasing [17]. In our case, the patient was stable with no severe symptoms and the condition of the intrauterine pregnancy was good. Moreover, the observed heterotopic mass in the left cornu of the uterus did not grow significantly throughout pregnancy. Therefore, expectant management was chosen. When expectant management is chosen, it is necessary to counsel patients about the possible complications and outcomes, regularly perform ultrasound examinations to assess the growth of ectopic pregnancy and intrauterine pregnancy and closely monitor the patient’s general condition. If expectant management is chosen, the risk of adverse outcomes is likely to be higher when the degree of asymmetry of the protrusion at the angle is high and the myometrium of the uterine angle is thin [18].

There is a shortage of medical literature regarding the different management techniques of heterotopic pregnancies and its results. A literature review by Kadjy et al. published in the 2021 reviewed 509 cases of heterotopic pregnancies. In 85 cases, the outcome of intrauterine pregnancy was described and in the 60% of these cases it ended with a live birth, however, these results were not distributed by the treatment strategy chosen [19].

Moreover, currently there are no guidelines for postpartum care for women with heterotopic pregnancy, if expectant management is chosen. In our case, ultrasound examinations were performed regularly after delivery and changes in blood β-hCG levels were observed for 1 month.

## 4. Conclusions

If intrauterine pregnancy is diagnosed, it is crucial to confirm that there is no additional extrauterine pregnancy, particularly in the presence of risk factors for heterotopic pregnancy. An MRI scan allows better visualization of soft tissues and helps to differentiate the heterotopic pregnancy in the corner of the uterus from the interstitial pregnancy. When choosing the treatment strategy for the heterotopic pregnancy, it is important to take into consideration the localization of the ectopic pregnancy, the size of the extrauterine gestation as well as its craniocaudal dimension and the presence of embryonic cardiac activity. When a heterotopic pregnancy is diagnosed, expectant management can be chosen. It may reduce the frequency of unnecessary interventions and help to prevent patients from complications associated with these interventions. If any indications occur or the condition of the patient or intrauterine pregnancy worsens, the expectant management should be changed to surgical or medical treatment.

## Figures and Tables

**Figure 1 medicina-57-01207-f001:**
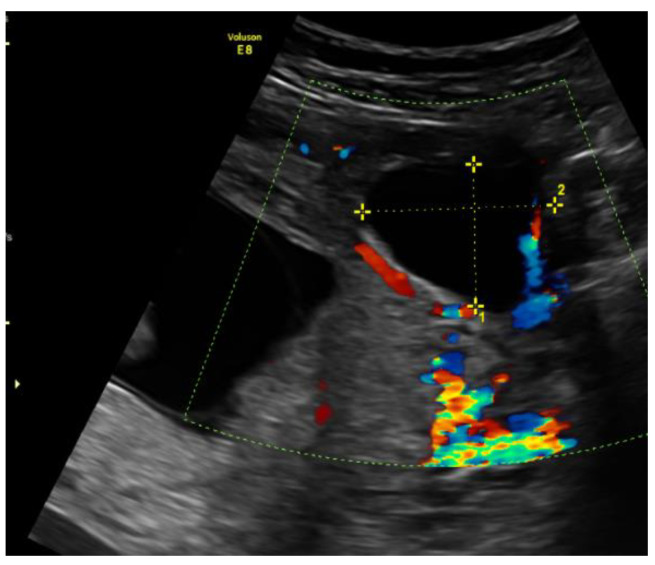
Left cornu of the uterus visualized by transabdominal ultrasonography on the 13w + 3d.

**Figure 2 medicina-57-01207-f002:**
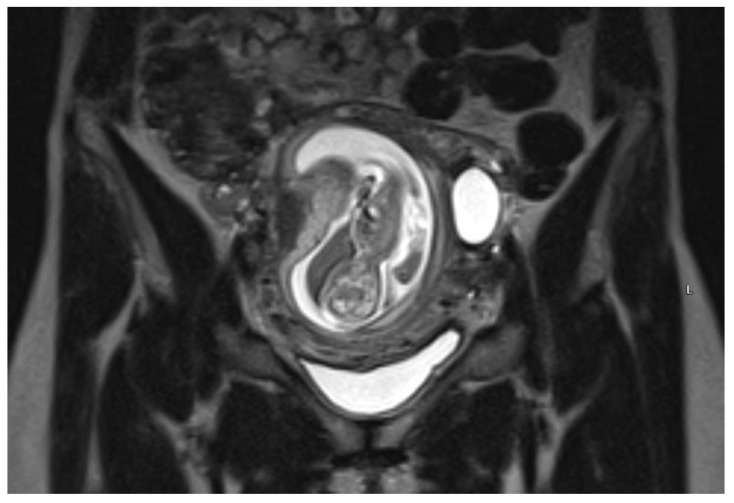
MRI scan of the uterus on the 13w + 3d.

**Figure 3 medicina-57-01207-f003:**
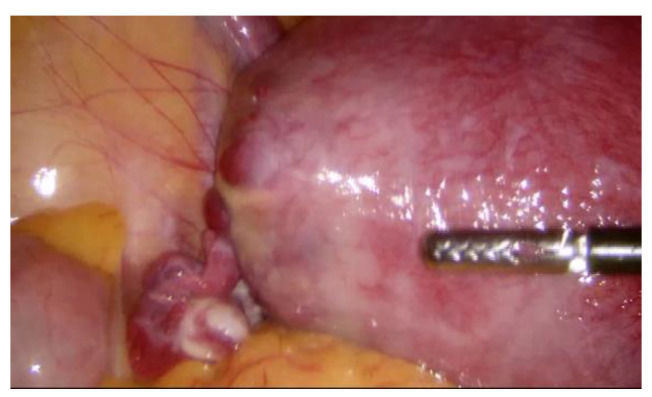
Left cornu of the uterus, observed during diagnostic laparoscopy.

**Figure 4 medicina-57-01207-f004:**
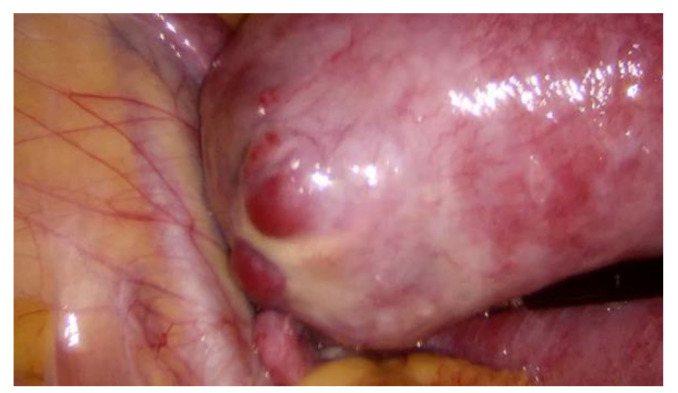
Left cornu of the uterus, observed during diagnostic laparoscopy.

**Figure 5 medicina-57-01207-f005:**
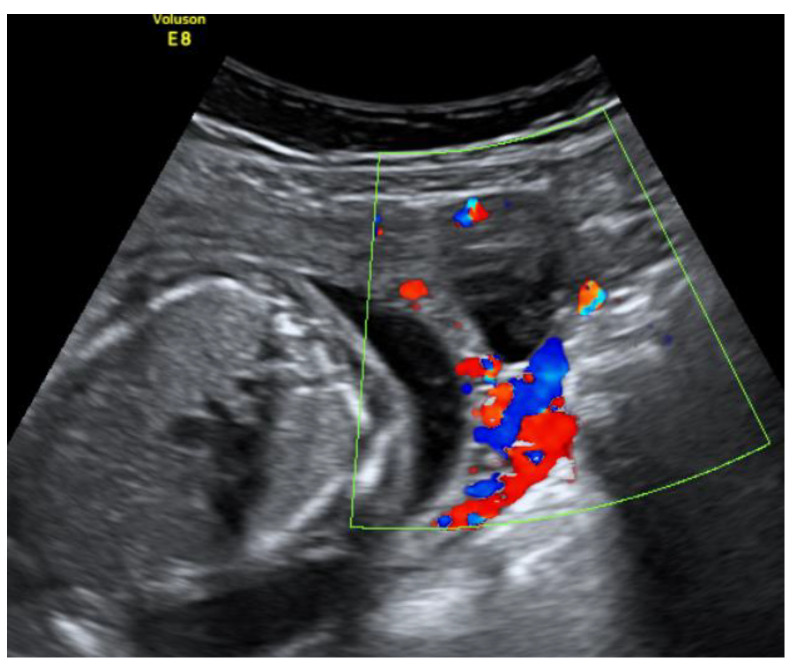
Left cornu of the uterus visualized by transabdominal ultrasonography at the 22nd week of gestation.

**Figure 6 medicina-57-01207-f006:**
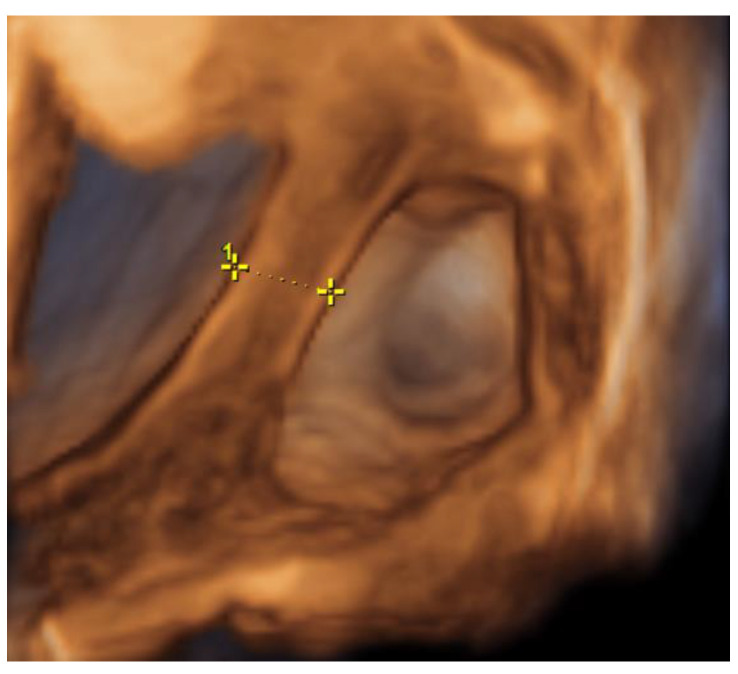
Left cornu of the uterus visualized by 3D transabdominal ultrasonography at the 22nd week of gestation.

**Figure 7 medicina-57-01207-f007:**
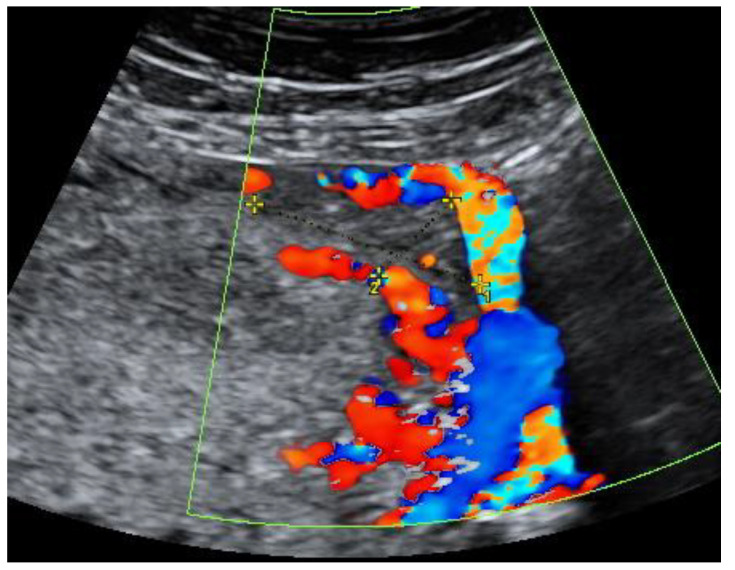
Left cornu of the uterus visualized by transabdominal ultrasonography 1 day postpartum.

**Figure 8 medicina-57-01207-f008:**
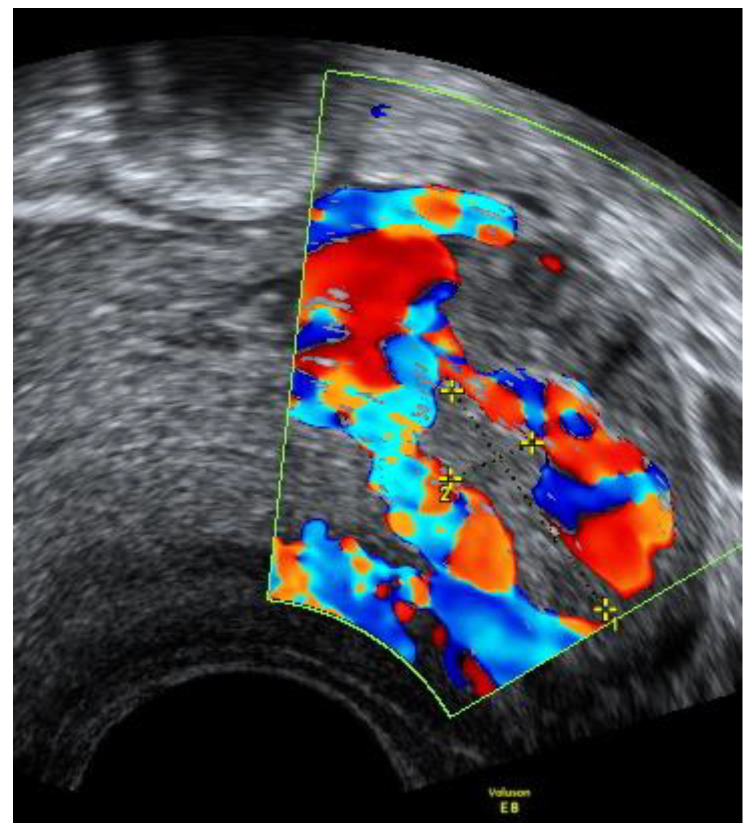
Left cornu of the uterus visualized by transvaginal ultrasonography 30 days postpartum.

**Table 1 medicina-57-01207-t001:** Patient’s β human chorionic gonadotropin (β-hCG) levels postpartum.

Days Postpartum	1	7	14	30
β-hCG (IU/L)	3602	59.4	4.82	1.78

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
