# Peer review of "Angular Heterotopic Pregnancy: Successful Differential Diagnosis, Expectant Management and Postpartum Care"

_medicina, 2021, doi:10.3390/medicina57111207_

Round 1

Reviewer 1 Report

This case report entitled "Angular Heterotopic Pregnancy: successful differential diagnosis, expectant management and postpartum care" describes an angular heterotopic pregnancy case in a 34-year- old patient and successful pregnancy outcome. 

The case report covers an important topic and is presented succinctly, however, a few additional details are needed for better presentation of the manuscript-

1- It's not clear if the patient content was needed and granted for publication.

2- The post-delivery follow-up is mentioned up to 30 days with significantly decreased β-hCG but the remaining mass with more intensive blood flow was still present. What happened to that mass after that? Any further details will be useful for future similar case management.

3- Starting lines of the abstract and introduction are the same. Needs changes to avoid repetition.

Reviewer 2 Report

The presented case of heterotopic pregnancy is interesting. Nevertheless, I would like to obtain some additional information, i.e.:

- What was the laparoscopic treatment in the previous pregnancy (probably in the third one)? At what gestational age was the operative procedure performed?

- What medications were used for "ovarian stimulation" in the current pregnancy?

- What was the course of pregnancy until the date of admission to the Perinatology Center (on the 14th week of gestation)? Was the diagnosis of heterotopic pregnancy suspected by the patient's outpatient physician?

- What recommendations did the patient receive during the last visit (i.e. one month after delivery), including the date of the further ultrasound visit, recommendations for potential procreation, etc.

Please attach a draft/figure that shows a localization of both pregnancies in the presented patient.

Please use "Instruction for Authors" for all references.
